# Side-group chemical gating via reversible optical and electric control in a single molecule transistor

Linan Meng[1,2,3], Na Xin[2], Chen Hu[4], Jinying Wang[2], Bo Gui[5], Junjie Shi[6], Cheng Wang[5], Cheng Shen[1], Guangyu Zhang[1], Hong Guo[4], Sheng Meng[1,3] & Xuefeng Guo [2,7]

By taking advantage of large changes in geometric and electronic structure during the reversible *trans–cis* isomerisation, azobenzene derivatives have been widely studied for potential applications in information processing and digital storage devices. Here we report an unusual discovery of unambiguous conductance switching upon light and electric field-induced isomerisation of azobenzene in a robust single-molecule electronic device for the first time. Both experimental and theoretical data consistently demonstrate that the azobenzene sidegroup serves as a viable chemical gate controlled by electric field, which efficiently modulates the energy difference of *trans* and *cis* forms as well as the energy barrier of isomerisation. In conjunction with photoinduced switching at low biases, these results afford a chemically-gateable, fully-reversible, two-mode, single-molecule transistor, offering a fresh perspective for creating future multifunctional single-molecule optoelectronic devices in a practical way.

[1] Institute of Physics, Chinese Academy of Sciences, Beijing 100190, PR China. [2] Beijing National Laboratory for Molecular Sciences, State Key Laboratory for Structural Chemistry of Unstable and Stable Species, College of Chemistry and Molecular Engineering, Peking University, Beijing 100871, PR China. [3] University of Chinese Academy of Sciences, Beijing 100049, PR China. [4] Center for the Physics of Materials and Department of Physics, McGill University, Montreal, QC H3A 2T8, Canada. [5] Key Laboratory of Biomedical Polymers of Ministry of Education, College of Chemistry and Molecular Sciences, Wuhan University, Wuhan 430072, PR China. [6] School of Chemistry & Chemical Engineering, Shandong University, Jinan 250100 Shandong, PR China. [7] Department of Materials Science and Engineering, College of Engineering, Peking University, Beijing 100871, PR China. These authors contributed equally: Linan Meng, Na Xin, Chen Hu. Correspondence and requests for materials should be addressed to H.G. (email: hong.guo@mcgill.ca) or to S.M. (email: smeng@iphy.ac.cn) or to X.G. (email: guoxf@pku.edu.cn)

Since the initial proposal of molecular electronics in the 1970s, single-molecule devices have been regarded as a promising solution to reach the ultimate limit of device miniaturisation[1]. To this end, researchers from interdisciplinary backgrounds have paid considerable efforts in studying the charge transport behaviours through molecular junctions[2]. To date, various functionalities have been achieved[2,3], which include switching[4,5], rectification[6–8], electrical field gating[9,10], mechanical force gating[11–14], thermoelectric energy conversion[15–17], electroluminescence[18], and magnetoresistance[19,20]. In addition to device functionalisation, single-molecule devices also provide a unique platform to explore the intrinsic properties of materials at the atomic/molecular level and reveal novel quantum phenomena that are inaccessible in bulky materials[21]. Among discrete functional devices, controllable electrical transistors are particularly attractive because they are the basic component in electrical circuits, and have been widely studied in the field of molecular electronics[22]. In general, there are several mechanisms to realise a reversible transition between two conductance states[23]: conformational changes (usually stimulated by light, ion, and junction elongation/compression[4,13,14]), valence-state changes controlled by an electrochemical potential[24], and spin-state changes often based on spin-crossover compounds[19,25,26]. In most cases, changes occur in the molecular backbone through which carriers are transported. It is worth mentioning that atoms or groups at the sidegroup position, which have even no continuous σ-bond with the junction, have been demonstrated to have significant effects on the electronic or geometric properties of the molecular backbones, as well as the conductance behaviours[27–29], forming the basis for creating functional molecular devices.

Azobenzene derivatives are typically photochromic materials, which are able to isomerise back-and-forth between two states (cis/trans) under irradiation of different wavelengths of light (UV/visible). This process can result in two significant changes in either the structure or the polarity of the azobenzene unit. One is that the trans conformation is near planar, while the cis isomer adopts a bent conformation with its phenyl rings twisted ~55° out of the plane from the azo group. The other is that the polarity change depends on the substituents on the two phenyl rings. In addition, the two isomers differentiate in the end-to-end distance[30]—the distance between the two carbon atoms in position 4 of the phenyl rings is ~9.0 and ~5.5 Å in trans form and cis form, respectively. These properties make azobenzene units popular in many systems, such as polymers[31], liquid crystals[32], and nanoparticles[33], which have produced various novel applications including digital storage[34] and on-command drug delivery[35]. At the molecular level, in addition to external light stimuli, previous reports based on scanning tunneling microscopy (STM) suggested that resonant or inelastic tunneling of electrons[36,37] or the electric field[38] could induce the isomerisation of azobenzene.

In the present work, we demonstrate a chemically-gated, fully-reversible, two-mode, single-molecule transistor based on a robust graphene-molecule-graphene single-molecule junction (GMG-SMJ) formed by covalently sandwiching an azobenzene unit, which is immobilised at the side position of a terphenyl aromatic chain, between graphene point contacts (Fig. 1a). One mode is asymmetric stochastic conductance switching due to electric field-induced isomerisation of azobenzene and the other is reversible photoswitching at low biases. Note that this azobenzene single-molecule transistor, which is switched between an ON and OFF state because the two isomers have the different resistances, is different from common single-molecule transistors in which the source-drain currents can be continuously tuned by using the gate voltage[9,39]. A crucial discovery is that both quantum mechanical calculations and experimental data reveal

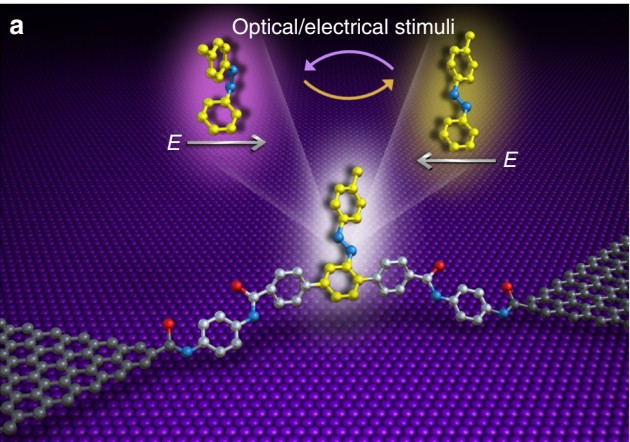

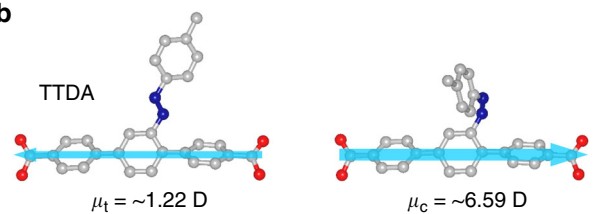

**Fig. 1** Device structure and dipole analysis of TTDA. **a** Schematic representation of the device structure that highlights a reversible isomerisation of the azobenzene unit between *trans* and *cis* forms triggered by either optical or electrical stimuli. The azobenzene sidegroup serves as a chemical gate to modulate the conductance of the molecular backbone. **b** Dipole projection on the molecular backbone (the actual charge transport pathway). The arrow denotes the direction of the dipole projection

that an electric field can unambiguously modulate the energy difference between *trans* and *cis* forms, as well as the energy barrier of conformational changes. Since the effectiveness of an electric field on *trans* and *cis* is different, either of the two conformations can be tuned as the stable state, which is the first study in single-molecule electrical circuits.

## Results

**Design and device fabrication.** The synthesis of 2′-p-tolyldiazenyl-1,1′:4,4′-terphenyl-4,4″-dicarboxylic acid (TTDA, Fig. 1b) was done according to a previous report[40]. Then, by using nanogapped graphene point contacts fabricated through a dash-line lithographic (DLL) method described elsewhere[41], we covalently incorporated individual TTDA molecules into graphene electrodes that have been prefunctionalised by p-phenylenediamine through amide bonds to form single-molecule junctions (Fig. 1, Supplementary Figures 1–3). Due to the covalent bond at the electrode–molecule contact interface and the good compatibility between graphene electrodes and molecules, these reconnected single-molecule junctions are stable, ensuring the following investigation of the stimuli-responsive conductance behaviours of the azobenzene unit.

As mentioned before, *trans* and *cis* forms of the azobenzene unit have distinct effects on the electronic structure of the whole molecule. In the current case, we chose a conjugated aromatic chain containing an azobenzene sidegroup as the molecular bridge because the isomerisation of azobenzene incorporated through side substitution does not essentially alter the geometric structure of the charge transport pathway, although there is a large geometric change of the azobenzene unit. In comparison with the cases where isomerisation groups are placed in the backbone, the present design avoids the possible instability of

molecular junctions resulting from the length change during isomerisation. This point is of crucial importance to achieve stable and reversible switching effects, and makes it more suitable to analyse the origin of the conductance differences between *trans* and *cis* forms on the basis of our previous unreported systematic exploration. Furthermore, we used the Gaussian package to investigate the electronic structure of these two isomers. As shown in Fig. 1b, the two forms of azobenzene show different dipole moments projecting an opposite direction along the terphenyl backbone (~1.22D and ~6.59D for *trans* and *cis* forms, respectively), revealing a significant asymmetry of the conformations. This different asymmetry between *trans* and *cis* forms is still preserved in our device system (Supplementary Figure 6). Previous reports have demonstrated that an electric field could act on asymmetry molecules, thus changing the conformation[42,43]. This strongly implies that other than the constant energy difference at the ground state, electric field could be utilised to tune the energy alignment of *trans* and *cis* forms as well as the transition energy barrier, resulting in an isomerisation of the substituted azobenzene unit in a single-molecule electrical circuit.

**Voltage-dependent stochastic switching**. On the basis of the investigation of current–voltage (I–V) characteristics of TTDA-based single-molecule junctions in the dark, asymmetric stochastic switching between two distinct conductance states were observed mainly at bias voltages in the range of −0.05 to −0.20 V, as represented in Fig. 2 out of 7 working devices. The I–V characteristics over the temperatures ranging from 115 to 250 K revealed that the switching rate between these two states is accelerated by increasing the temperature. We note that this voltage-dependent switching behaviour is obviously distinguishable from stochastic switching induced by the conformational

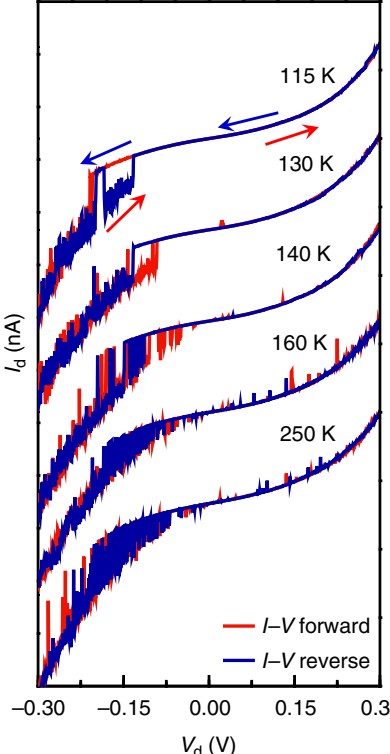

**Fig. 2** *I–V* characteristics of TTDA molecular junctions at different temperatures. The devices were measured in the dark and under vacuum. Stochastic switching mainly occurred at the negative bias voltages. The gate voltage is 0 V

change of the hexaphenyl aromatic chain previously reported (different dihedral angles between the outer two benzene rings)[44]. The asymmetrical voltage-dependence and small threshold voltage values suggest that the charge-trap mechanism cannot explain the present stochastic switching phenomenon[45].

To investigate such voltage dependence in detail, we performed real-time current measurements at different bias voltages ranging from −0.05 to −0.30 V in the dark and under vacuum. The obtained current–time (I–t) curves and the corresponding conductance-based histograms at 160 K are shown in Fig. 3. It is clear that with increasing the negative voltages from −0.05 to −0.30 V, the proportion of the high conductance state grew markedly, whereas the proportion of the low conductance state gradually decreased to zero. On the basis of I–t data, the on/off ratios of the conductance switching were calculated to be in the range from ~2.00 to ~2.46, showing little dependence on bias voltages (Supplementary Table 1). To rule out potential artifacts, we also investigated the conductance behaviour of a control molecule—a pure terphenyl aromatic chain without any substituted azobenzene sidegroup (Supplementary Figures 4, 5). There existed only one single conductance state. We did not observe the same stochastic switching at similar temperatures (Supplementary Figure 11) and bias voltages (Supplementary Figure 12). Collectively, we believe that the stochastic switching effect in the azobenzene–terphenyl aromatic chain originates from the azobenzene unit, other than the terphenyl unit due to δ-bond-rotation-induced stereoelectronics[44].

**Photoswitching effect at low bias voltages**. We attributed the two conductance states to corresponding molecular conformations by further revealing the photoresponsive behaviour of the azobenzene unit. As shown in Fig. 4, the device reversibly switched its conductance between a high state and a low state with an on/off ratio of ~2.1 under sequential ultraviolet (UV) and visible light irradiations, where a very small positive voltage of 0.01 V was adopted to prevent the effect of electric field. The similar on/off ratio with stochastic switching in Fig. 3 suggests the same isomerisation process. Control experiments by using a device reconnected by a terphenyl aromatic molecule without the substituted azobenzene sidegroup (Supplementary Figure 4) did not show similar photoswitching behaviours (Supplementary Figure 13), thus proving that the photoswitching effect as well as electric field-induced stochastic switching should originate from reversible *trans–cis* isomerisation of azobenzene. The large current fluctuation under UV irradiation might be due to the photoinduced instability of *cis* azobenzene or some other unknown stereoelectronic effect, which deserves further studies in the future.

**Discussion**
To reveal how the different isomers of azobenzene affect the conductance behaviour of the terphenyl aromatic chain, we analysed the equilibrium transmission spectra of both *trans* and *cis* GMG-SMJs by using density functional theory (DFT)-based nonequilibrium Green's function (NEGF) approach. As shown in Fig. 5a, in comparison with the *trans* form, the perturbed frontier molecular orbitals (p-FMOs) of the *cis* form shift positively in the energy window. This is because the *cis* possesses poorer conjugation within the side azobenzene structure than the *trans* (see also the scattering state analysis in Supplementary Figure 7). Based on the Landauer–Büttiker formalism of quantum transport, the energy gap between p-FMOs and the Fermi level of the electrodes plays a crucial role in the current under a low bias voltage: the smaller the gap is, the more electrons can enter the Fermi window in the bias range explored. As the perturbed

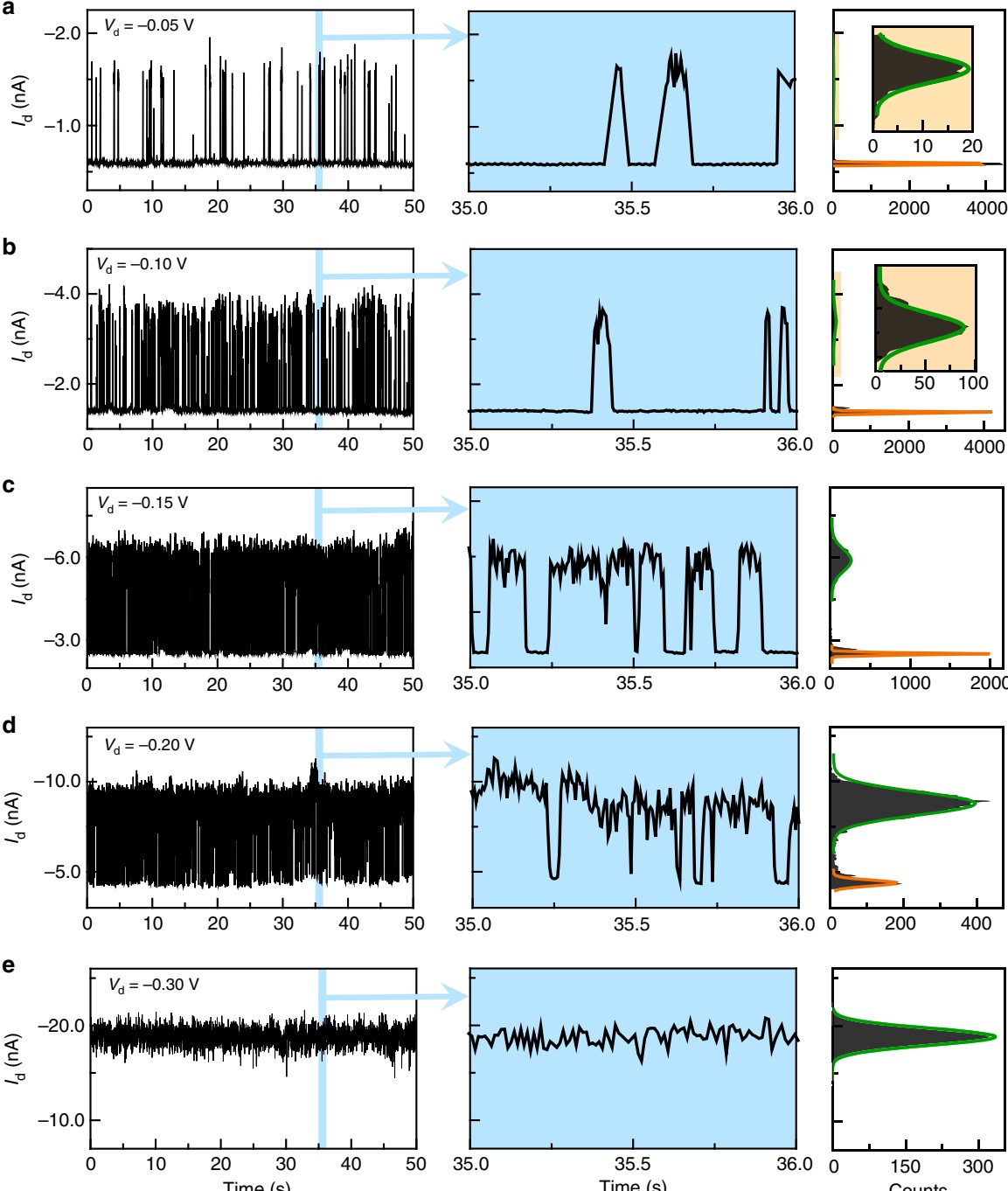

**Fig. 3** Voltage dependence of stochastic switching. **a–e** I–t curves, corresponding enlarge I–t curves marked in blue, and corresponding histograms of a working GMG-SMJ device measured at different bias voltages: −0.05 V (**a**), −0.10 V (**b**), −0.15 V (**c**), −0.20 V (**d**), and −0.30 V (**e**) at 160 K. Insets in the right column are the enlarged curves marked in yellow. The gate voltage is 0 V

highest occupied molecular orbital (p-HOMO) is much closer to the graphene Fermi level than the perturbed lowest unoccupied molecular orbital (p-LUMO), p-HOMO provides the dominant contribution to the conductance at small bias voltages. When the azobenzene changes from *trans* to *cis*, p-HOMO moves towards the graphene Fermi level (~0.07 eV), thus leading to larger transmission around the Fermi level and higher conductance at small bias voltages. Therefore, we theoretically attributed low and high conductance states to *trans* and *cis* forms, respectively, which is consistent with the experimentally-observed photo-switching effect in Fig. 4. It should be mentioned that there is a

discrepancy between the observed conductances and the predicted transmission.

Next, we investigated the experimentally observed process of switching from low to high conductance states by increasing the voltage from −0.05 to −0.30 V in the dark (Fig. 3). We shall reach the conclusion that applying negative voltages can isomerise azobenzene from *trans* to *cis* gradually.

Given the fact that the proportions of *trans* and *cis* forms can be tuned directly by the voltage in the dark, a modulated thermodynamic isomerisation energy is $\Delta G = -RT\ln([H]/[L])$, where $[H]$ and $[L]$ are the occurrence proportions of *cis* and *trans* forms,

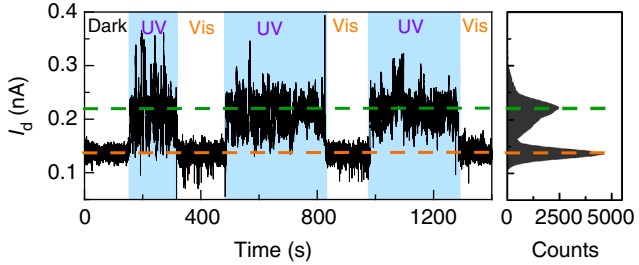

**Fig. 4** Photoswitching behaviours. At low bias voltages, real-time measurements reveal reversible switching between *cis* and *trans* forms of TTDA upon exposure to UV and visible light irradiations, respectively. The right panel is the corresponding statistic histogram, showing that the conductance of *cis* form is higher than that of *trans* form. The source-drain voltage is 0.01 V; the gate voltage is 0 V

respectively (Supplementary Table 2). As shown in Fig. 5b, $-\ln([H]/[L])$ changes linearly with bias voltages and decreased to zero when negative voltages increased to $-0.13$ V. This implies a changed energy landscape between *trans* and *cis* under the electric field. To confirm the intrinsic mechanism of this asymmetric voltage-dependent *trans–cis* isomerisation, we performed first-principles calculations on the energy evolution of *trans* and *cis* forms of azobenzene in the GMG-SMJ system. We globally define the isomerisation energy as the energy difference between *cis* and *trans* forms: $\Delta E = E^c - E^t$, where the superscript $c$ and $t$ denote *cis* and *trans*, respectively. In a biased two-probe system, $\Delta E$ is determined by two factors: one is the intrinsic isomerisation energy in the absence of voltages, $\Delta E_0 = E_0^c - E_0^t$; and the other is the voltage-induced electrostatic energy difference defined as $\Delta\varepsilon = \varepsilon^c - \varepsilon^t$. The intrinsic isomerisation energy is around 70 kJ mol$^{-1}$ (the energy value at zero bias as shown in Fig. 5c), resulting in a favourable state of *trans* at the equilibrium condition. The

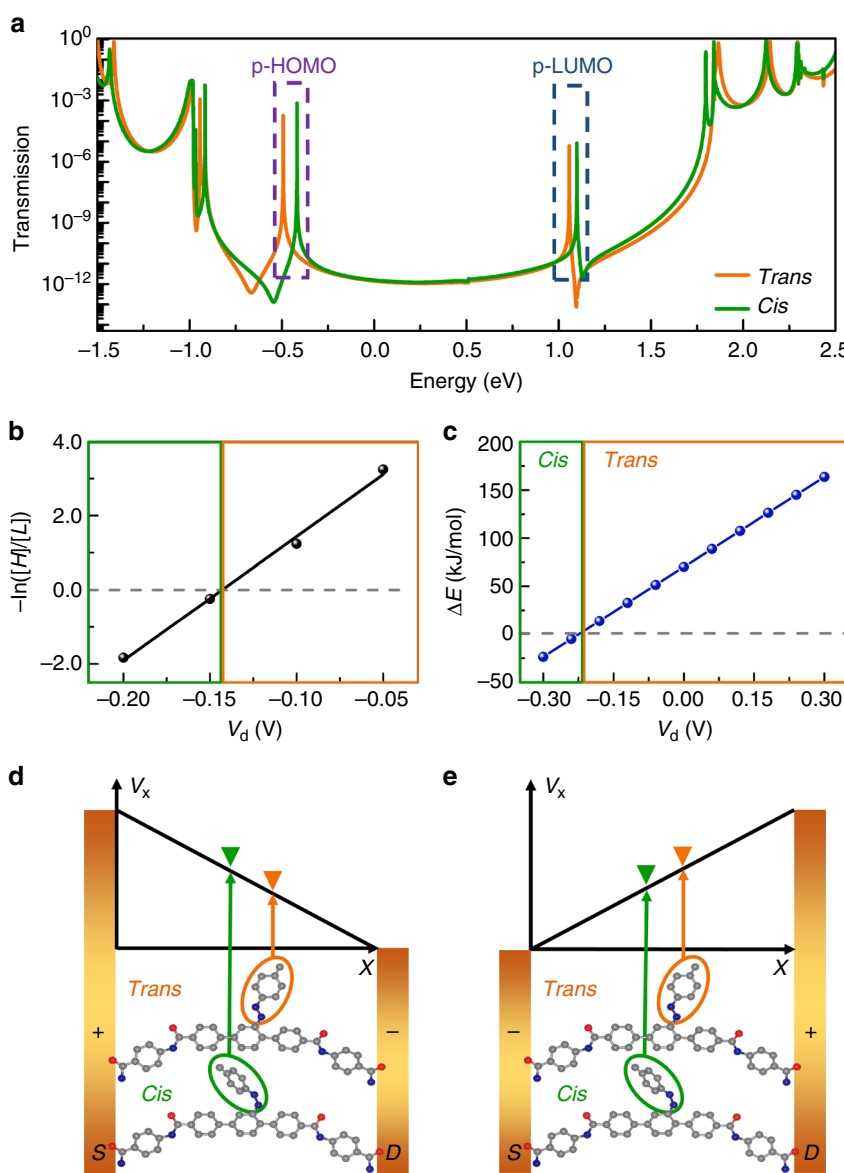

**Fig. 5** Mechanism of voltage-dependent conductance switching. **a** Transmission spectra of a GMG-SMJ with *trans* (orange) and *cis* (green) isomers, respectively. **b** Voltage-dependent thermodynamic isomerisation energy parameters deduced from experimental data. **c** Voltage-dependent isomerisation energy diagram, which is calculated from the theoretical simulations. **d**, **e** Schematic of relative electric potential energies of *trans* and *cis* forms under positive and negative bias electric fields, respectively. The voltage drops of *trans* and *cis* systems were approximately treated as the same linear tendency $V_x$ for a simple sketch demonstration

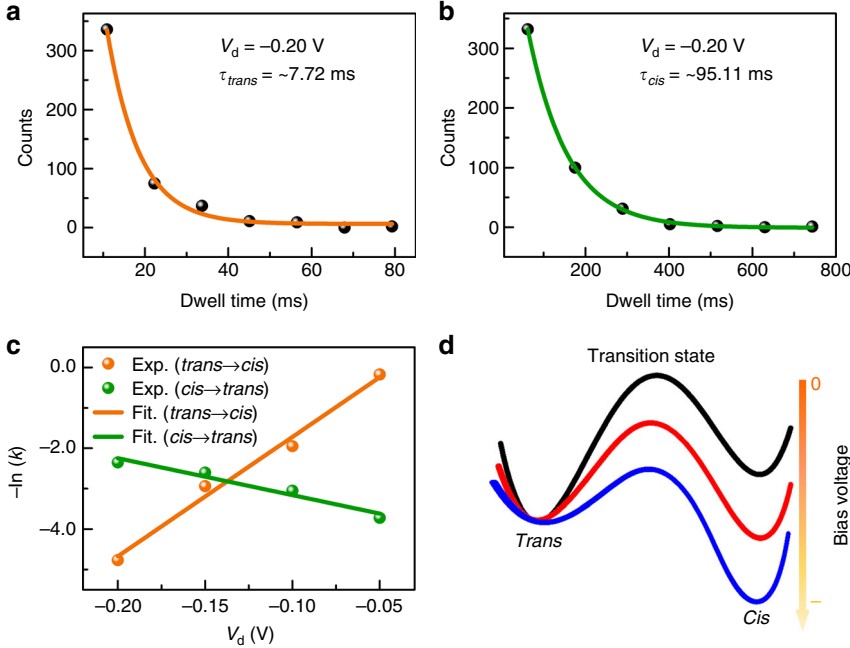

**Fig. 6** Relationship between the activation energy and the bias voltage. **a**, **b** Statistics histograms of the dwell times at *trans* (**a**) and *cis* (**b**) states fitted by a single-exponential function. **c** Activation energy parameters of *trans–cis* (orange) and *cis–trans* (green) as a function of the bias voltage. The variation of *cis–trans* is less than *trans–cis*. **d** Schematic of the energy alignment among *trans*, *cis*, and transition states triggered by the bias voltage. We take the energy of *trans* as the reference

voltage-induced electrostatic energy difference $\Delta\varepsilon$ of the two-probe junction under different biases were calculated by the DFT–NEGF technique and obtained by the formula: $\Delta\varepsilon = -|e|\int \left[ \rho_c(r)V_c(r) - \rho_t(r)V_t(r) \right] dr$, where e is the electron charge, $\rho(r)$ and $V(r)$ are the self-consistently-calculated electron density distribution and electric potential distributions induced by the bias electric field at the molecular device region. Indices $c$ and $t$ denote *cis* and *trans* conformation systems, respectively. The calculated function between $\Delta\varepsilon$ and voltages was shown in Supplementary Figure 8, where the $\Delta\varepsilon$ value is found to be linearly dependent on the bias voltages. We combined the bias-dependent $\Delta\varepsilon$ with the zero-bias $\Delta E_0$ to obtain the total under-bias isomerisation energy $\Delta E$, and the energy diagram was shown in Fig. 5c. With the help of a negative voltage, the isomerisation energy from *trans* to *cis* decreased linearly and reached the critical transition point ($\Delta E = 0$) at −0.21 V, since $\Delta\varepsilon$ can totally offset the intrinsic molecular energy difference under this bias condition. When the bias goes more negative, *cis* is the more stable state than *trans* ($\Delta E < 0$), according to the energy diagram in Fig. 5c. In contrast, at the positive voltage direction, the energy difference between *cis* and *trans* forms maintains the rising trend along with the voltage, resulting in a continued stable state of *trans*. Both the transition voltage and linear changing tendency are qualitatively consistent with experimental data of the thermodynamic energy term $-\ln([H]/[L])$ in Fig. 5b. On account of the fixed spatial position of the backbone, the main electron density difference of the *trans* and *cis* structures results from the azobenzene free phenyl ring (side group marked by the orange and green circles, respectively), which dominates $\Delta\varepsilon$ under biases as shown in the schematic demonstrations of Fig. 5d, e. In addition, this physical picture revealed that structural asymmetries of *trans* and *cis* play the key role in the response to the applied bias. Considering the randomness of the molecular-device fabrication, it can be inferred that an opposite-orientation-aligned molecule should have the opposite polarity of the bias response: the positive bias deceased the energy difference and the transition

point appeared at the positive bias region. This inference was confirmed by six other devices as shown in Supplementary Figure 9, demonstrating the robustness and reproducibility. Collectively, both experimental and theoretical analyses consistently establish the physical picture that in the two-probe junction, the mechanism of stochastic switching is the capability of an externally applied electric field to adjust the molecular energies of *trans* and *cis* forms that determine the most stable state. It is the different forms of azobenzene (either *trans* or *cis*) that serve as an efficient chemical gate (because of their different dipole moments as discussed before) to tune charge transport of the molecular main chain.

We further investigate the dynamic process of activating the molecule to overcome the energy barrier between *cis* and *trans* forms. On the basis of Figs. 2 and 3, we observed two facts. One is that the stochastic jumping occurs immediately as long as a small negative voltage is applied. The other is that the isomerisation rate gradually increases with increasing the voltages. To unravel the kinetics of azobenzene isomerisation induced by electric field, the I–t curves were idealised based on the hidden-Markov-model simulation by using a QUB software (Supplementary Figure 10), from which the time intervals of *trans* and *cis* forms ($T_{trans}$ and $T_{cis}$) can be obtained. Plots of the dwell times of each form obtained by frequency analysis for a typical signal at a bias of −0.20 V are shown in Fig. 6a, b. Both plots fit a single-exponential function with time constants corresponding to the average lifetimes ($\tau_{trans}$ and $\tau_{cis}$) of *trans* and *cis* forms. The average lifetimes of *trans* and *cis* forms at 160 K under different bias voltages were obtained as listed in Supplementary Table 2. With increasing the negative voltage, $\tau_{trans}$ decreases from ~842.37 to ~8.50 ms, while $\tau_{cis}$ increases from ~24.13 to ~95.11 ms. Accordingly, the rates of isomerisation $k_{t\rightarrow c} = 1/\tau_{trans}$ (from *trans* to *cis*) and $k_{c\rightarrow t} = 1/\tau_{cis}$ (from *cis* to *trans*) have the opposite changing tendency. For the isomerisation rate, there exists an expression as $-\ln k \propto E_a/RT$, where $E_a$ is the activation energy (a free energy difference between the stable state and the transition

state). Here, we can define $E_a^{t \to c} = E_{a,0}^{t \to c} + \varepsilon_a^{t \to c} = E_{a,0}^{t \to c} + (\varepsilon^T - \varepsilon^t)$ and $E_a^{c \to t} = E_{a,0}^{c \to t} + \varepsilon_a^{c \to t} = E_{a,0}^{c \to t} + (\varepsilon^T - \varepsilon^c)$, where the superscript T denotes the transition state during the isomerisation process and $E_{a,0}$ is the intrinsic activation energy barrier without the bias field effect. In this way, $\varepsilon^T - \varepsilon^t$ and $\varepsilon^T - \varepsilon^c$ determine how $E_a^{t \to c}$ and $E_a^{c \to t}$ change upon the application of the bias voltage, respectively. As shown in Fig. 6c, both $-\ln k_{t \to c}$ and $-\ln k_{c \to t}$ depend linearly on the negative voltage but with an opposite changing trend and the different slopes in the numeric value. Therefore, with increasing the negative voltage, $\varepsilon^T - \varepsilon^t$ decreases and $\varepsilon^T - \varepsilon^c$ increases, which means that the additional energy of the transition state caused by electric field is between those of *trans* and *cis* forms. This is also consistent with the model shown in Fig. 5d, e as the position of the azobenzene sidegroup in the transition state is between those of *trans* and *cis* forms. Figure 6d qualitatively shows the schematic voltage-dependent energy diagram of the molecules in *trans*, *cis*, and transition states, where the energy of the *trans* form is set as the reference. We can infer that the appearance of stochastic jumping is a result of the tunable activation energy motivated by thermal or inelastic tunneling of electrons, or their cooperative effect. It is worth mentioning that at a small voltage of 0.01 V, the structure of azobenzene can be stabilised (either *cis* and *trans*) for long-term measurements as demonstrated in Supplementary Figure 14, thus producing a resettable single-molecule memoriser.

This work demonstrated a new phenomenon of asymmetric stochastic conductance switching via efficient side-group chemical gating in a GMG-SMJ-based electrical circuit. The key to the success in realising such a switching effect is the efficient adjustment of the energy alignments among *trans*, *cis*, and transition states by an external electric field, which plays a crucial role in the *trans*–*cis* isomerisation process, due to the asymmetric spatial orientation of the azobenzene unit. In combination with the photoswitching phenomenon, these devices behave as a chemically-gated, fully-reversible, two-mode, single-molecule molecular circuit, thus providing new insights into constructing future practical single-molecule devices and logic gates.

## Methods

**Device fabrication and molecular connection**. High-quality single-layer graphene grown through a chemical vapor deposition (CVD) process was first transferred to the silicon wafer with a 300 nm silicon oxide layer. Then the source/drain metal electrode arrays (8 nm Cr/60 nm Au) were patterned by photolithography and thermal evaporation. The devices with carboxylic acid-terminated nanogapped graphene point contacts (the top panel in Supplementary Figures 2 and 4) were fabricated by a DLL method described in detail elsewhere[41]. Individual TTDA molecules were connected to graphene point contacts by a dehydration reaction in two steps (Supplementary Figures 2 and 4). In brief, in the first step, *p*-phenylenediamine was dissolved in anhydrous pyridine with the concentration of about $10^{-3}$ M. Then, the freshly-cut graphene devices and 1-ethyl-3-(3-dimethylaminopropyl) carbodiimide hydrochloride (EDCI), a well-known carbodiimide dehydrating/activating agent, were added to the solution for connection. After 2 days in the dark and argon atmosphere, the devices were taken out from the solution, cleaned sequentially by ultrapure water and acetone, and dried by nitrogen gas. In the second step, the resulted devices and EDCI were added to the solution of TTDA with the concentration of $10^{-4}$ M. After 2 days in the dark and argon atmosphere, the devices were taken out from the solution, cleaned sequentially by ultrapure water and acetone, and finally dried by nitrogen gas. In this work, TTDA was synthesised by following the reported procedure[40], and the controlled molecule (4-[4-(4-carboxyphenyl)phenyl]benzoic acid) was purchased from Accela Chem Bio Co. Ltd. and used without further purification.

**Electrical characterisation**. Device characterisations were carried out by utilising an Agilent 4155C semiconductor characterisation system and a ST-500-Probe station (Janis Research Company) with a liquid nitrogen cooling system. For *I–V* measurements, the scanning interval is 0.6 mV per step. For *I–t* measurements, the integration time is 5 ms per step. UV irradiation ($\lambda = 365$ nm) was carried out with a handheld lamp (CU6, NITECORE). The visible light was obtained from a polychromatic light with a UV light filter (420 nm cut-off wavelength).

**Theoretical calculation**. The structural relaxation of the isolated molecules and two-probe transport junctions, the molecular dipole, and the intrinsic molecular energy calculation were carried out by the DFT in the Gaussian package[46]. The B3LYP hybrid functional and the 6-31G basis set were taken in the Gaussian calculation. The electric potential energy under finite biases and the charge transport properties of the two-probe structures were obtained by carrying out the DFT within the NEGF formalism, as implemented in the quantum transport package Nanodcal[47,48]. Double-zeta polarised atomic orbital basis set was used in the NEGF–DFT calculation, and exchange-correlation was treated at the PBE-GGA level. The cutoff energy for the real-space grid was set at 1360 eV. The NEGF–DFT self-consistent calculations were deemed converged when every element of the Hamiltonian matrix and the density matrix were converged to less than $10^{-4}$ a.u, and the total energy was converged to less than $10^{-5}$ a.u. In the transmission spectra (Fig. 5a), the Fermi level of the graphene electrode ($E_f$) has been shifted $-0.5$ eV to simulate the experimental fact that our graphene electrode is p-doped due to the wet transfer procedure. The transmission spectra and scattering states were calculated by the Green's function and scattering matrix methods (Supplementary Figure 7). As shown in Supplementary Figure 7, p-HOMO and p-LUMO are contributed from the orbitals around the branch chain. The *trans* form has a better conjugated structure around the branch chain than the *cis* form, leading to lower p-FMOs energies.

## Data availability

The data that support the findings of this study are available from the corresponding authors upon request.

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

## Acknowledgements

The authors acknowledge primary financial supports from National Key R&D Program of China (2017YFA0204901 and 2016YFA0300902), the National Natural Science Foundation of China (Grants 21727806, 11774396 and 11474328), the Natural Science Foundation of Beijing (Z181100004418003), the Natural Sciences and Engineering Research Council of Canada (H.G.), and the Interdisciplinary Medicine Seed Fund of Peking University. The authors thank the CalcuQuebec and Compute Canada for computation facilities.

## Author contributions

X.G., S.M. and H.G. conceived and designed the experiments. L.M. and N.X. fabricated the devices. L.M., N.X., C.S. and G.Z. performed the device measurements. B.G., J.S. and C.W. carried out the molecular synthesis. C.H. and H.G. built and analysed the theoretical bias-induced isomerisation model and performed the quantum transport calculation. C.H. and J.W. performed the isolated molecule calculation. X.G., H.G., S.M., L.M., N.X. and C.H. analysed the data and wrote the paper. All the authors discussed the results and commented on the manuscript.

## Additional information

**Competing interests:** The authors declare no competing interests.

