## [Peer Review File · Nature Communications]

Reviewers' comments:

Reviewer #1 (Remarks to the Author):

Referee report

Title: Side-Group Chemical Gating via Reversible Optical and Electric Control in a Single-Molecule Transistor

By L. Meng et al.

This is a combined experimental and theoretical manuscript on a single molecule transistor. Gating of the molecule occurs via a side-group which can be switched between two states via an optical or electrical signal. The molecule is switched between its trans (planar) to cis (bent) configuration or vice versa. Since the two configurations have different resistances (high/low) the molecule can be switched between an ON and an OFF state.

The single molecule devices are fabricated by using photolithography and electron beam lithography. The molecules are connected to graphene point contacts via a dehydration reaction process.

The paper is solid and scientifically sound. In addition, the manuscript is very well written and has excellent figures. The results are, as far as I can judge, original. The storyline of the manuscript is easy to follow and the experimental results are, as far as I can judge, valid. As experimentalist I'm unable to judge on the quality of the theoretical calculations.

Remarks:

- 1) Lines 57-58. Please add that molecules can also be mechanically gated, i.e. by compressing or stretching a molecule its conductance can be systematically varied.
- 2) The molecule is switched between its trans (planar) to cis (bent) configuration or vice versa. Since the two configurations have different resistances (high/low) the molecule can be switched between an ON and an OFF state. I would like to emphasize that this single molecule 'device' differs from a conventional transistor where the source-drain current can be tuned continuously varied (using the gate voltage). I encourage the authors to explicitly mention this difference between their single molecule transistor and a conventional transistor.
- 3) I can imagine that the interaction between the graphene contacts and the molecules is not always the same and that it can also be affected by the applied electric field. Why are the authors so confident that contact effects can be ignored in their work?

In summary, I find this an interesting manuscript reporting important and novel findings that will probably be of interest to a relatively large community. After satisfactory revision the manuscript might be suited for publication in Nature Communications.

Reviewer #2 (Remarks to the Author):

In the manuscript titled "Side-Group Chemical Gating via Reversible Optical and Electric Control in a Single-Molecule Transistor", the authors report controlling a conductance of a single-molecule junction using a polyphenyl molecule with an azobenzene unit, which is synthesized by them. It is a unique idea to introduce the azobenzene unit as a side group of a molecular wire, which does not change the bridging length but can affect the conductance through a change in the dipole moment. The single-

molecule device is fabricated on graphene nanoelectrodes that have also been originally developed by the authors. Conductance switching of the molecular junction through a conformational change of the azobenzene unit is induced either by an electric field in the junction or by light irradiation. Under the electric field, the azobenzene unit switches between two states, most probably trans and cis, in a reversible manner, which is observed only at one bias polarity. Under irradiation, the conductance (molecular conformation) can be reversibly switched by UV and visible light. The simulations provide mechanistic insight into the reactions.

I found that authors demonstrate a new possibility to control electron transportation in a molecular junction which is, in general, important in nanoscale science. However, I do not see a clear perspective how this molecule could work as a single-molecule transistor that the authors claim. The electric field induced switching is a stochastic process and seems impossible to keep a specific state (e.g., to store information or perform calculations). In addition, the on-off ratio of the conductance switching is very small and it is not clear what kind of task the molecule could do with it.

Concerning the mechanism of the azobenzene switching induced by an electric field or light, it has been well-studied using low-temperature scanning tunneling microscopy, which has provided detailed information at the single-molecule level with high-quality images of individual molecules (e.g., Ref. 34-36, more examples: PRL 99, 038301 (2007), Angew. Chem. Int. Ed. 57, 15034 (2018), ACS Nano 12, 1821 (2018)). Although the authors claim "In STM studies, however, the substrates have non-negligible effect on the behavior of adsorbates through physical/chemical interactions, which might lead to a variety of influencing factors and the difficulty for precise control.", such interactions between a molecule and a substrate should be of fundamental importance in device applications. Even in the present case, the chemical interaction (bonding) of the molecule with the graphene electrode should play a crucial role to determine its electric properties and switching dynamics. In addition, the molecule may also interact with the device substrate. The authors provide a reasonable explanation of the switching mechanism, a novel aspect is rather limited.

The manuscript is well-written with high-quality of experimental data and theoretical simulations, which definitely deserve to be published. However, I'm not convinced that the manuscript meets all criteria that are required for Nature Communications.

Reviewer #3 (Remarks to the Author):

Review of Nature Communications manuscript NCOMMS 18 33116A Z

The field of molecular electronics has been greatly invigorated by the development of reliable methods for making and characterizing metal-single molecule-metal junctions in the last 15 years. Methods for externally 'gating' molecules so that these junctions can be used as electrical switches is of obvious interest; electrostatic 'gating' (using a third electrode in the solid state, or by electrochemical control using an electrolyte solution) is one route, but a photochemical approach where the molecule is 'switched' by light has potential advantages. Use of graphene, instead of traditional metals, as the contacts offers less possibility of quenching molecular excited states in photochemical switching experiments.

This paper reports such a study, in which an azobenzene derivative is connected between two graphene contacts and its conductance switches depending upon whether the azobenzene is in the cis or trans form. The data presented do indeed suggest a molecular effect because a control molecule in which the azobenzene unit is omitted does not show switching. Given the current state of the art in

this field (graphene contact experiments are still challenging, and it seems that few groups can yet routinely perform such studies) and the possible significance of the results, this paper might meet the Journal criteria for publication, but only if the authors can satisfactorily address the following significant technical issues before publication.

The discussion in this paper, as in previous papers from this group, consistently refers to 'single molecule junctions'. However, it is not at all clear how the reader can be sure that single molecule junctions are indeed formed, for the following reason. Papers from this group (including the present paper) all reference a 2012 *Angewandte* paper (ref. 38 here) as a full account of their methodology (incidentally, this reference is incorrectly cited; it should be volume 51, pages 12228-12232). In the latter paper, Figure 2 illustrates the device fabrication procedure. The devices include multiple pairs of metal contacts (Figure 2d). Let us assume that between each pair of metal contacts, there is a parallel track of graphene point contacts made as indicated in figure 2e (the experimental description in the paper and in the SI is not unambiguous on this point). Given that each pair of metal contacts overlaps by about 50 microns, and that the total length occupied by one graphene point contact pair is 190 nm (figure 2a), there would be about $50000/190$, or about 260 graphene point contact pairs addressed by each pair of source and drain electrodes. What are the chances that, on average, only one of those point contacts is properly connected, by a single molecule, and that this is moreover repeated consistently for all the devices on the chip (that is, all of the pairs of parallel metal source and drain electrodes)?

Looking further at the data in the SI for this 2012 *Angewandte* paper, the authors publish some tables giving conductances for multiple junctions made by re-connecting broken graphene point contacts with a series of 3 molecules. Taking just the first example, where an oligo(phenyl ether)sulfone derivative was employed, the conductances at low bias range from $0.4G_0$ to $10^4 G_0$. This does not seem consistent with the reliable formation of single molecule junctions, especially given the occurrence of such high conductance values for 'single molecule junctions' involving a nonconjugated oligo(phenyl ether)sulfone.

From the above, it should be evident that it is not straightforward for the reader to deduce how the devices are formed or to assess the validity of the methods in the present paper. In fact, very little is said about how reliably the devices showing the claimed switching phenomena are formed. On P7 of the manuscript we are told that Fig. 2 shows data typical of '7 working devices'. But how many devices were made and tested? What proportion show the claimed behavior? What do the devices show that do not have the claimed behavior? This is nowhere discussed. The history of molecular electronics is unfortunately replete with papers claiming remarkable physics, but seen on a single junction (or a small handful of junctions), but if other workers cannot reproduce it, it is of limited value.

In Figure 3 we are presented with data from a single device, and the expected random orientation of molecules is supported by presenting characteristics for another device in the SI Figure S7, but this is two devices! Out of how many? Again, if only 7 devices were observed in total, surely the data for all seven should be included, even if only in the SI.

The authors should include in the SI (if not in the main paper) a detailed appraisal of the properties of the full range of junctions typically formed with these molecules (both the azo derivative and the control), including calculated conductances as in their earlier *Angewandte* paper, and give statistics for successful vs failed junction formation. Assuming that some junctions successfully form but do not show switching, these results should be included too.

Only then would it be possible to assess the likely significance of the main conclusions from this work.

Reply to Reviewer 1:

Comments: This is a combined experimental and theoretical manuscript on a single molecule transistor. Gating of the molecule occurs via a side-group which can be switched between two states via an optical or electrical signal. The molecule is switched between its *trans* (planar) to *cis* (bent) configuration or vice versa. Since the two configurations have different resistances (high/low) the molecule can be switched between an ON and an OFF state. The single molecule devices are fabricated by using photolithography and electron beam lithography. The molecules are connected to graphene point contacts via a dehydration reaction process. The paper is solid and scientifically sound. In addition, the manuscript is very well written and has excellent figures. The results are, as far as I can judge, original. The storyline of the manuscript is easy to follow and the experimental results are, as far as I can judge, valid. As experimentalist I'm unable to judge on the quality of the theoretical calculations.

We thank the referee very much for his/her high evaluation and kind recommendation. As for the theoretical calculation, the results were from our state-of-the-art first principles approach that accounts for all the microscopic details of the devices from an atomistic point of view, and the numerical accuracy was carefully verified by convergence tests regarding to control parameters such as real space and k-space meshes, basis sets, etc.

Comment 1: Lines 57-58. Please add that molecules can also be mechanically gated, i.e. by compressing or stretching a molecule its conductance can be systematically varied.

Answer: Thanks for the good suggestion. We have added “mechanical force gating” in Line 57 accordingly and cited the corresponding references as Refs. 11-14 (*Nat. Nanotechnol.* **2009**, *4*, 230; *Nat. Nanotechnol.* **2012**, *7*, 35; *Nat. Nanotechnol.* **2013**, *8*, 282; *Nat. Chem.* **2015**, *7*, 215).

Comment 2: The molecule is switched between its *trans* (planar) to *cis* (bent) configuration or vice versa. Since the two configurations have different resistances (high/low) the molecule can be switched between an ON and an OFF state. I would like to emphasize that this single molecule ‘device’ differs from a conventional transistor where the source-drain current can be tuned

continuously varied (using the gate voltage). I encourage the authors to explicitly mention this difference between their single molecule transistor and a conventional transistor.

Answer: Thanks for the helpful suggestion. We agree with the referee's comment. The mechanisms for conventional transistors and single-molecule transistors are totally different although the source-drain currents of both can be continuously tuned by using a gate voltage (Ref. 9). In a conventional transistor, the carrier density is modulated by the gate voltage while in a single-molecule transistor, the energy gap between the dominant conducting molecular orbital energy level and Fermi level of electrodes is tuned by the gate voltage (*Angew. Chem. Int. Ed.* **2018**, *57*, 14026). In the current case of azobenzene single-molecule transistors, the conformation-induced switching mechanism we presented is also different from the general single-molecule transistor. To clearly explain this, we added the comment in the main manuscript (Lines 97-100) as follows: "Note that this azobenzene single-molecule transistor, which is switched between an ON and OFF state because the two isomers have the different resistances, is different from common single-molecule transistors in which the source-drain currents can be continuously tuned by using the gate voltage (*Chem. Soc. Rev.* **2015**, *44*, 902; *Angew. Chem. Int. Ed.* **2018**, *57*, 14026)."

Comment 3: I can imagine that the interaction between the graphene contacts and the molecules is not always the same and that it can also be affected by the applied electric field. Why are the authors so confident that contact effects can be ignored in their work?

Answer: Thanks for the comment. We would like to take this opportunity to provide our explanation. The contact effects can be excluded as the reason for the observed conductance variations in this work for the following reasons. Firstly, the ON/OFF ratio of the two conductance states in Fig. 2 is consistent with that of the photo-induced conductance switching, which is measured under a very small bias voltage (~10 mV). Secondly, we observed only one conductance state in the control molecular system of a pure terphenyl aromatic chain (Figs. S9-S11), where the anchor groups and the contact details are the same with that of the azobenzene-terphenyl chain. The results of the control group provide a direct and solid evidence that the contact details have no observable effect on the two-modes ON/OFF switching reported here.

In summary, I find this an interesting manuscript reporting important and novel findings that will probably be of interest to a relatively large community. After satisfactory revision the manuscript might be suited for publication in *Nature Communications*.

Answer: Thanks very much for the high evaluation. After careful revision, we believe that the revised manuscript is now suitable for publication in *Nature Communications*.

Reply to Reviewer 2:

Comments: In the manuscript titled “Side-Group Chemical Gating via Reversible Optical and Electric Control in a Single-Molecule Transistor”, the authors report controlling a conductance of a single-molecule junction using a polyphenyl molecule with an azobenzene unit, which is synthesized by them. It is a unique idea to introduce the azobenzene unit as a side group of a molecular wire, which does not change the bridging length but can affect the conductance through a change in the dipole moment. The single-molecule device is fabricated on graphene nanoelectrodes that have also been originally developed by the authors. Conductance switching of the molecular junction through a conformational change of the azobenzene unit is induced either by an electric field in the junction or by light irradiation. Under the electric field, the azobenzene unit switches between two states, most probably *trans* and *cis*, in a reversible manner, which is observed only at one bias polarity. Under irradiation, the conductance (molecular conformation) can be reversibly switched by UV and visible light. The simulations provide mechanistic insight into the reactions.

We thank the referee for his/her high evaluation and kind recommendation.

Comment 1: I found that authors demonstrate a new possibility to control electron transportation in a molecular junction which is, in general, important in nanoscale science. However, I do not see a clear perspective how this molecule could work as a single-molecule transistor that the authors claim. The electric field induced switching is a stochastic process and seems impossible to keep a specific state (e.g., to storage information or perform calculations). In addition, the on-off ratio of the conductance switching is very small and it is not clear what kind of task the molecule could do with it.

Answer: We would like to take this opportunity to explain the functions of our azobenzene single-molecule transistors. As shown in Figure 4, the devices can be switched in two *stable* conductance states (low/high) by UV and visible lights, which is useful for storing optical information or performing calculations. In addition to photoswitching, the field-induced switching can be also utilized to store information. For example, as shown in Fig. S12, the large voltages (0.30 V or -0.70 V) were applied to trigger the *trans-cis* isomerization (information writing), and a small voltage ~ 0.01 V was applied to read the ON/OFF state.

We admit that the ON/OFF ratio of the conductance switching is small in the current case. We would like to emphasize that in the present work, we aimed to deliver the design concept of how to build a new type of chemically-gated single-molecule transistors and then develop the prototype device. This concept is very important to make judicious device design through flexible molecular engineering for improving the performance, which will attract intense attention.

Comment 2: Concerning the mechanism of the azobenzene switching induced by an electric field or light, it has been well-studied using low-temperature scanning tunneling microscopy, which has provided detailed information at the single-molecule level with high-quality images of individual molecules (e.g., Ref. 34-36, more examples: PRL 99, 038301 (2007), Angew. Chem. Int. Ed. 57, 15034 (2018), ACS Nano 12, 1821 (2018)). Although the authors claim “In STM studies, however, the substrates have non-negligible effect on the behavior of adsorbates through physical/chemical interactions, which might lead to a variety of influencing factors and the difficulty for precise control.”, such interactions between a molecule and a substrate should be of fundamental importance in device applications. Even in the present case, the chemical interaction (bonding) of the molecule with the graphene electrode should play a crucial role to determine its electric properties and switching dynamics. In addition, the molecule may also interact with the device substrate. The authors provide a reasonable explanation of the switching mechanism, a novel aspect is rather limited.

Answer: Thanks for the comment. We agree with the referee about the fact that the interaction between a molecule and a substrate is of fundamental importance in device applications. Systematic investigations have been made in the mechanism of the azobenzene switching in the presence of electric field or light. In general, azobenzene adsorption on metal surfaces typically results in quenching of the light-induced isomerization, owing to its strong electronic coupling to

the substrate. To realize light-induced switching, several strategies have been adopted: 1) adding bulky side groups to the azobenzene unit to reduce the interaction with the surface (*Phys. Rev. Lett.* **2007**, 99, 038301); 2) integrating the azobenzene unit into the molecular structures that are inherently rigid and three dimensional, such as the tripod structures (*Angew. Chem. Int. Ed.* **2018**, 57, 15304; *J. Phys. Chem. B* **2006**, 110, 1968); 3) using bulk insulators as the supporting substrate (*ACS Nano* **2018**, 12, 1821). In addition, resonant or inelastic tunneling electrons (Refs. 34-35) or the electric field (Ref. 36) could induce the isomerization of azobenzene.

In our present case, the photoswitching phenomenon demonstrates that the electronic coupling between the azobenzene unit and graphene electrodes is weak, which is consistent with the interface coupling strength (proportional to the peak width of transmission) for both the *trans* form and the *cis* form obtained from the calculate transmission spectra as shown in Fig. 5a. The device substrate is a silicon wafer with a 300 nm silicon oxide layer, which is insulating. According to previous works (*Phys. Rev. Lett.* **2010**, 104, 216102; *ACS Nano* **2018**, 12, 1821), the substrate effect could be ignored in the two-terminal measurement.

We also would like to take this opportunity to detail the comparison between our present work and the STM system as follows. In our case, an electric field between source and drain electrodes can efficiently adjust the energy alignments among *trans*, *cis* and transition states, which plays a crucial role in the *trans*–*cis* isomerization process, due to the asymmetric spatial orientation of the azobenzene unit. As a result, we observed an asymmetric stochastic conductance switching phenomenon due to *trans*-*cis* isomerization, and realized memory functionality. This fundamental mechanism and practical applications are totally different from the previous work based on the STM system (*J. Am. Chem. Soc.* **2006**, 128, 14446), which also demonstrated electric field-induced azobenzene isomerization because the electric field can deform the potential energy surface related to the reaction and lead to an effective lowering of the isomerization barrier. As long as the electric field is high enough, the isomerization both from *trans* to *cis* and from *cis* to *trans* can be observed, which is uncontrollable and do not promise further applications.

The manuscript is well-written with high-quality of experimental data and theoretical simulations, which definitely deserve to be published. However, I'm not convinced that the manuscript meets all criteria that are required for Nature Communications.

Answer: Thanks very much for the high evaluation. After revision, we believe that the revised manuscript is now suitable for publication in *Nature Communications*.

Reply to Reviewer 3:

Comments: The field of molecular electronics has been greatly invigorated by the development of reliable methods for making and characterizing metal-single molecule-metal junctions in the last 15 years. Methods for externally ‘gating’ molecules so that these junctions can be used as electrical switches is of obvious interest; electrostatic ‘gating’ (using a third electrode in the solid state, or by electrochemical control using an electrolyte solution) is one route, but a photochemical approach where the molecule is ‘switched’ by light has potential advantages. Use of graphene, instead of traditional metals, as the contacts offers less possibility of quenching molecular excited states in photochemical switching experiments. This paper reports such a study, in which an azobenzene derivative is connected between two graphene contacts and its conductance switches depending upon whether the azobenzene is in the cis or trans form. The data presented do indeed suggest a molecular effect because a control molecule in which the azobenzene unit is omitted does not show switching. Given the current state of the art in this field (graphene contact experiments are still challenging, and it seems that few groups can yet routinely perform such studies) and the possible significance of the results, this paper might meet the Journal criteria for publication, but only if the authors can satisfactorily address the following significant technical issues before publication.

Thank the referee very much for his/her high evaluation and kind recommendation.

Comment 1: The discussion in this paper, as in previous papers from this group, consistently refers to ‘single molecule junctions’. However, it is not at all clear how the reader can be sure that single molecule junctions are indeed formed, for the following reason. Papers from this group (including the present paper) all reference a 2012 *Angewandte* paper (ref. 38 here) as a full account of their methodology (incidentally, this reference is incorrectly cited; it should be volume 51, pages 12228-12232). In the latter paper, Figure 2 illustrates the device fabrication procedure. The devices include multiple pairs of metal contacts (Figure 2d). Let us assume that between each pair of metal contacts, there is a parallel track of graphene point contacts made as indicated in figure 2e (the

experimental description in the paper and in the SI is not unambiguous on this point). Given that each pair of metal contacts overlaps by about 50 microns, and that the total length occupied by one graphene point contact pair is 190 nm (figure 2a), there would be about 50000/190, or about 260 graphene point contact pairs addressed by each pair of source and drain electrodes. What are the chances that, on average, only one of those point contacts is properly connected, by a single molecule, and that this is moreover repeated consistently for all the devices on the chip (that is, all of the pairs of parallel metal source and drain electrodes)?

Answer: Thanks for the insightful comment. According to the suggestion, the original Ref. 38 are revised as “41. Cao, Y. *et al.* Building high-throughput molecular junctions using indented graphene point contacts. *Angew. Chem. Int. Ed.* **51**, 12238-12232 (2012).”

The analysis of single-molecule connection has been provided in Page S5 of the Supplementary Information (SI). In our device, the length of graphene among two pair of metal electrodes is 40 microns. Since the distance of graphene point contacts is 190 nm (Ref. 41), there are about 210 (40000/190) graphene point contact pairs (junctions) between source and drain electrodes. The possibility of rejoined device with n junctions can be addressed:

$$G_n = \frac{m!}{n!(m-n)!} p^n (1-p)^{m-n}$$

Where m is the number of graphene point contact pairs (210), n is the connected contact pairs and p is the possibility of successful connection for a contact pair. In our experiments, ~5% devices show successful connection of molecules and the corresponding possibility of successful connection for each contact pairs (p) is $0.05/m = 0.024\%$. Therefore, the possibility of connected device γ_c can be attained:

$$\gamma_c = 1 - G_0 = 1 - \frac{m!}{0!(m-0)!} p^0 (1-p)^m = 1 - (1-p)^m$$

Where G_0 is the possibility of device without any reconnected junctions. $\gamma_c = 5\%$ can achieved taking $p = 0.024\%$. Then, the ratio of single-junction devices to the overall reconnected devices [G_1/γ_c] is ~97.5%, which suggests that, in most cases, charge transport in these devices arises mainly in a single-molecule junction.

In addition, in our previous work (please see *Science* **2016**, 352, 1443), we used the same method to fabricate the devices. In that work, we performed the systematic low-temperature measurements, especially inelastic electron tunneling spectra (IETS) (Figures S9 and S11), which proved that only a single molecule was covalently anchored onto graphene point contacts.

Comment 2: Looking further at the data in the SI for this 2012 *Angewandte* paper, the authors publish some tables giving conductances for multiple junctions made by re-connecting broken graphene point contacts with a series of 3 molecules. Taking just the first example, where an oligo(phenyl ether)sulfone derivative was employed, the conductances at low bias range from $0.4G_0$ to $10^{-4}G_0$. This does not seem consistent with the reliable formation of single molecule junctions, especially given the occurrence of such high conductance values for ‘single molecule junctions’ involving a nonconjugated oligo(phenyl ether)sulfone.

Answer: Thanks for the comment. One advantage of the “graphene-molecule-graphene” single-molecule junctions developed by our group is that the contact between single molecules and graphene electrode is amido covalent bond, which significantly improves the stability of the devices and promises the formation of robust single-molecule junctions, which can endure chemical treatments and external stimuli. The conductance variance of single-molecule junctions has always been a challenge in the fields of nanoscale devices or molecular devices, which should be overcome for the vision of practical molecular electronic devices. As the device size decreases down to the atomic/molecular levels, every atomic details are able to affect the device performance. In our graphene-based single-molecule devices, the realization of atomic level precision in the cutting procedure to obtain graphene nanogaps, precise control of the molecular conformation on the substrate within the graphene gaps, and the contact configuration are still the existing challenges. These factors contribute to the variability among devices featuring the same structure of single molecules. Although the conductance varies from device to device, their functions in the current case are reliable and reproducible, as demonstrated by the switching effects in the current case.

Considering the electronic structure of oligo(phenyl ether)sulfone molecule, electron donor groups (phenyl ring) and electron acceptor groups (oxygen and sulfone) are connected intimately with $p-\pi$ or $\pi-\pi$ conjugations, which enhance the electron delocalization and narrow the HOMO-LUMO gap, thus contributing to a high conductance value.

Comment 3: From the above, it should be evident that it is not straightforward for the reader to deduce how the devices are formed or to assess the validity of the methods in the present paper. In fact, very little is said about how reliably the devices showing the claimed switching phenomena are formed. On P7 of the manuscript we are told that Fig. 2 shows data typical of ‘7 working

devices'. But how many devices were made and tested? What proportion show the claimed behavior? What do the devices show that do not have the claimed behavior? This is nowhere discussed. The history of molecular electronics is unfortunately replete with papers claiming remarkable physics, but seen on a single junction (or a small handful of junctions), but if other workers cannot reproduce it, it is of limited value.

Answer: Thanks for the professional comment. We added the detailed description about how to fabricate the devices in the “Method” part of the main manuscript, which can make it straightforward for the reader to deduce how the devices are formed and to assess the validity of the methods in the present paper.

In the present manuscript, the possibility of successful connected junctions was about 5% due to the use of a two-step connection procedure (Page S5 in the Supplementary Information). For all the connected junctions, only devices with high conductance ($>10^{-4} G_0$ at ± 0.5 V) were measured. We have totally measured 9 devices, among which 7 devices showed the electric-field induced switching phenomenon (in addition to the one shown in the main manuscript, the *I-V* curves of the other 6 working devices have been added in Figure S7), and 2 devices failed to response to the electric-field trigger. We think that it might be the uncontrollable contact configuration that results in a strong interface coupling and hinders the isomerization between *trans* and *cis*.

Comment 4: In Figure 3 we are presented with data from a single device, and the expected random orientation of molecules is supported by presenting characteristics for another device in the SI Figure S7, but this is two devices! Out of how many? Again, if only 7 devices were observed in total, surely the data for all seven should be included, even if only in the SI.

Answer: Thanks for the helpful suggestion. We have added *I-V* characteristics of all the other 6 working devices in Figure S7. As shown in Figure S7 and Figure 2, there are 3 devices demonstrating the electric-field induced switching under negative bias voltages. For the other 4 devices, the stochastic switching occurred under positive bias voltages. We have also changed the corresponding description (Line 277 at Page 15) from “This inference was well confirmed by **another device** in Supplementary Figure S7” to “This inference was confirmed by **six other devices** as shown in Supplementary Figure S7”.

Comment 5: The authors should include in the SI (if not in the main paper) a detailed appraisal of the properties of the full range of junctions typically formed with these molecules (both the azo derivative and the control), including calculated conductances as in their earlier *Angewandte* paper, and give statistics for successful vs failed junction formation. Assuming that some junctions successfully form but do not show switching, these results should be included too.

Answer: Thanks for the helpful suggestion. The conductance statistics of measured DDTA single-molecule junctions (Table S3) and control terphenyl single-molecule junctions (Table S4) were added in the revised Supplementary Information.

Only then would it be possible to assess the likely significance of the main conclusions from this work.

Answer: Thanks very much for the high evaluation. After careful revision fully according to the referee's suggestions, we believe that the revised manuscript is now suitable for publication in *Nature Communications*.

REVIEWERS' COMMENTS:

Reviewer #1 (Remarks to the Author):

The authors have properly addressed all the points that I have raised in my previous report and the manuscript has been modified accordingly. I recommend publication of this manuscript in its present form.

Reviewer #2 (Remarks to the Author):

The authors answered to my question in a satisfactory manner. However, I still wonder why they want to emphasize a difference between their approach and the STM experiments with somewhat negative remarks for the latter. It is obvious that these are a very different technical approach and address different fundamental aspects of single-molecule devices. Therefore, I do not believe that some remarks by the authors are not very fair and hope that they reconsider them. Otherwise, the manuscript is very well-written and their conclusions are supported by the presented experimental and theoretical data. Now I recommend it to be published in Nature Communications.

Reviewer #3 (Remarks to the Author):

Report on paper NCOMMS 18 33116B

I thank the authors for their additional explanation of their reasoning in the reply, which was helpful in that I now understand the statistical basis for their claims for single molecule devices. Additionally they have accommodated some of the points I made in their revised version, and I think the addition of the other device characteristics and the tables of conductances to the supplementary information improves the paper significantly.

However, I am still very unconvinced by the evidence presented for the nature of the 'single molecule junctions'. In the present case, looking at the transmission functions for the purported device structure, shown in Figure 5a, the value of $T(E)$ near the Fermi (or should that be Dirac point?) energy is 10^{-12} , so the zero bias conductance expected for this system is $10^{-12} G_0$. Yet the lowest device conductance obtained by the authors is around $10^{-4} G_0$. Even allowing for all the factors involved in going from the Landauer formula (assumes zero bias, zero K) to 'real life', that is a staggering difference in conductance. Such a low predicted transmission is not unreasonable given that the authors' purported molecular system consists of five benzene rings and no fewer than four conjugation-breaking amide linkers! Incidentally, the authors claim in their reply that '...the electronic structure of oligo(phenyl ether)sulfone molecule, electron donor groups (phenyl ring) and electron acceptor groups (oxygen and sulfone) are connected intimately... which enhance the electron delocalization and narrow the HOMO-LUMO gap, thus contributing to a high conductance value.' But the analogous compound 4,4'-bis(phenoxy)diphenyl sulfone is described as a 'while crystalline solid' (see J. Mater. Chem. 1991, 1, 271), meaning that it is certainly not a low bandgap material; its HOMO-LUMO separation must be at least 3 eV.

Given the evidence, I do accept that something 'molecular' is going on in this system, and I accept that the other two referees were more favorably disposed towards the paper. So on those grounds I would not object to the publication of the paper, but I think that a cautionary note should be added somewhere in the discussion about the discrepancy between the observed conductances and the

predicted transmission.

Reply to Reviewer 1:

Comments: The authors have properly addressed all the points that I have raised in my previous report and the manuscript has been modified accordingly. I recommend publication of this manuscript in its present form.

Answer: We thank the referee very much for his/her high evaluation and kind recommendation.

Reply to Reviewer 2:

Comments: The authors answered to my question in a satisfactory manner. However, I still wonder why they want to emphasize a difference between their approach and the STM experiments with somewhat negative remarks for the latter. It is obvious that these are a very different technical approach and address different fundamental aspects of single-molecule devices. Therefore, I do not believe that some remarks by the authors are not very fair and hope that they reconsider them. Otherwise, the manuscript is very well-written and their conclusions are supported by the presented experimental and theoretical data. Now I recommend it to be published in Nature Communications.

Answer: We thank the referee very much for his/her high evaluation and kind recommendation. We agree with the referee's comment about the fact that "STM is a very different technical approach and address different fundamental aspects of single-molecule devices" and removed the discussion of STM in Page 4, paragraph 1.

Reply to Reviewer 3:

Comments: Report on paper NCOMMS 18 33116B

I thank the authors for their additional explanation of their reasoning in the reply, which was helpful in that I now understand the statistical basis for their claims for single molecule devices.

Additionally, they have accommodated some of the points I made in their revised version, and I think the addition of the other device characteristics and the tables of conductances to the supplementary information improves the paper significantly.

However, I am still very unconvinced by the evidence presented for the nature of the ‘single molecule junctions’. In the present case, looking at the transmission functions for the purported device structure, shown in Figure 5a, the value of $T(E)$ near the Fermi (or should that be Dirac point?) energy is 10^{-12} , so the zero bias conductance expected for this system is $10^{-12}G_0$. Yet the lowest device conductance obtained by the authors is around $10^{-4}G_0$. Even allowing for all the factors involved in going from the Landauer formula (assumes zero bias, zero K) to ‘real life’, that is a staggering difference in conductance. Such a low predicted transmission is not unreasonable given that the authors’ purported molecular system consists of five benzene rings and no fewer than four conjugation-breaking amide linkers! Incidentally, the authors claim in their reply that ‘...the electronic structure of oligo(phenyl ether)sulfone molecule, electron donor groups (phenyl ring) and electron acceptor groups (oxygen and sulfone) are connected intimately... which enhance the electron delocalization and narrow the HOMO-LUMO gap, thus contributing to a high conductance value.’ But the analogous compound 4,4'-bis(phenoxy)diphenyl sulfone is described as a ‘white crystalline solid’ (see *J. Mater. Chem.* 1991, 1, 271), meaning that it is certainly not a low bandgap material; its HOMO-LUMO separation must be at least 3 eV.

Given the evidence, I do accept that something ‘molecular’ is going on in this system, and I accept that the other two referees were more favorably disposed towards the paper. So on those grounds I would not object to the publication of the paper, but I think that a cautionary note should be added somewhere in the discussion about the discrepancy between the observed conductances and the predicted transmission.

Answer: We thank the referee very much for his/her high evaluation and kind recommendation. We agree with the referee’s comment about the discrepancy between the observed conductances and the predicted transmission and added a comment accordingly. Please see Lines 216-217 in Page 12 of the revised Manuscript as follows: “It should be mentioned that there is the discrepancy between the observed conductances and the predicted transmission.”